# The Outcomes of Liver Transplantation in Highly Dependent Incapacitated Patients with Intellectual and Developmental Disabilities

**DOI:** 10.3390/jcm13195702

**Published:** 2024-09-25

**Authors:** Michal Skalski, Oskar Kornasiewicz, Joanna Raszeja-Wyszomirska, Agata Konieczka, Monika Mlynarczyk, Michal Grat

**Affiliations:** Department of General, Transplant and Liver Surgery, Public Central Teaching Hospital, Medical University of Warsaw, 1A Banacha St., 02-097 Warsaw, Poland; oskar.kornasiewicz@wum.edu.pl (O.K.); joanna.wyszomirska@wum.edu.pl (J.R.-W.); agata.anna.konieczka@gmail.com (A.K.); monika.mlynarczyk@wum.edu.pl (M.M.); michal.grat@wum.edu.pl (M.G.)

**Keywords:** liver transplantation, incapacitated patient, intellectual disability, developmental disability

## Abstract

**Background/Objectives:** Data regarding the outcomes of liver transplantation in disabled, highly dependent, and legally incapacitated adults are scarce, likely due to the infrequency of these procedures in such populations. Multicenter studies in adult transplant centers have shown that patients with coexisting intellectual and developmental disabilities (IDDs) may be denied transplantation because of their expected low longevity and the complexities associated with managing post-transplant care. We examined the long-term patient and graft outcomes in highly dependent, incapacitated patients with IDDs who underwent elective transplantation for chronic liver disease. **Methods:** Six adult patients who underwent liver transplantation for primary biliary cholangitis (*n* = 2), hepatitis C cirrhosis (*n* = 2), Wilson’s disease (*n* = 1), and autoimmune hepatitis (*n* = 1) were included. The main causes of their disability were infantile cerebral palsy, myotonia, and Niemann–Pick disease. **Results:** Four of the six patients were women, with a median age of 26 (range: 23–36) years. Only one patient died during follow-up. Their 1- and 5-year survival rates were 100 and 75%, respectively, which were not statistically different from those of the general cohort of electively transplanted patients (95.8 and 90.1%, respectively) (*p* = 0.35). **Conclusions:** Adult patients who are highly dependent, disabled, or legally incapable should not be denied liver transplantation because of poor long-term survival rates. Physiological disorders and psychiatric comorbidities should not prevent patients from receiving life-saving surgeries due to poor postoperative compliance or low quality of life.

## 1. Introduction

There has recently been an increase in interest in liver transplantation for adult patients who are disabled, extremely dependent, and often legally incapacitated due to intellectual and developmental disabilities (IDDs). This may owe to the limited number of studies and reports on adults with intellectual disabilities receiving liver transplantation, as opposed to the pediatric population [1]. These adult patients mostly include individuals with cerebral palsy and congenital metabolic or genetic disorders. Adult patients who are highly dependent and legally incapacitated are rarely considered candidates for liver transplantation due to misconceptions about their ability for postoperative compliance. Although liver transplantation in these patients can be a challenging and complex procedure, it is a life-saving and enhancing treatment. Surveys in adult transplant centers have shown that patients may be denied transplantation because coexisting IDDs are considered contraindications [2]. Potentially inequitable access to deceased donor organs for transplantation is a serious issue that may disproportionately affect patients with impaired decision-making capacities. This means that individuals who are unable to make their own decisions regarding transplantation may be less likely to receive a transplant than those who are able [3]. To address this potential problem, the European Society of Transplantation established an “Ethical and Legal Issues” working group to initiate the expert consensus process [3]. The decision-making process for liver transplantation includes both medical and psychosocial factors, considering that transplantation is likely to provide significant health benefits, although no specific policy has been developed for this patient population [4]. More cases of solid organ transplantation in patients with IDDs need to be reported to gain a broader perspective on this issue before definitive conclusions are drawn. To our knowledge, studies on this subject are scarce, and only limited data are available regarding liver transplantation in this specific patient cohort. Recently, successful liver transplantation in a highly dependent patient with Down’s syndrome was reported [5].

In this study, we aimed to investigate the consequences of transplantations in highly dependent, incapacitated adult patients in terms of their long-term outcomes and their effect on caregivers’ quality of life (QoL).

## 2. Materials and Methods

We conducted a retrospective cohort analysis of a prospectively collected database of highly dependent incapacitated patients who received liver transplants from donation after brain death (DBD) donors among the 2717 liver transplants performed in the Department of General, Transplant and Liver Surgery of the Public Central Teaching Hospital, the Medical University of Warsaw, between 2000 and 2022.

Legally incapacitated adult patients were defined as those without the capacity to make appropriate decisions. By court order, all their decisions, particularly life-related ones, were delegated to a caregiver appointed by the family court. All the participants had altered consciousness, and in some cases, there was evidence of self- or environmental awareness, such as simple commands, gestures, responses, intelligible verbalization, and purposeful behavior.

Six patients who fulfilled the aforementioned criteria were compared with transplanted adult patients with chronic liver disease and no disabilities, who were selected using propensity score matching based on age.

The six patients included had undergone elective liver transplantation for primary biliary cholangitis (*n* = 2), hepatitis C cirrhosis (*n* = 2), Wilson’s disease (*n* = 1), or autoimmune hepatitis (*n* = 1). Their main conditions resulting in disability were infantile cerebral palsy (*n* = 3), myotonia (*n* = 1), and Niemann–Pick disease (*n* = 2). Their demographical and clinical data are presented in Table 1.

This article was reviewed by the bioethics committee of Warsaw Medical University.

Kaplan–Meier survival analysis was performed to assess patient survival. Between-group differences in survival were assessed using the log-rank test. Statistical significance was set at *p* < 0.05.

All the statistical analyses were performed using the R language via RStudio Team (2020) (RStudio: Integrated Development for RStudio, PBC, Boston, MA, USA; URL http://rstudio.com/) (R version 4.3.1 (16 June 2023)).

## 3. Results

Highly dependent and incapacitated patients receiving liver transplants are extremely rare on transplant waiting lists. They accounted for <0.2% of all the transplant recipients in our cohort. The medical and social conditions of the patients was carefully analyzed at a multidisciplinary meeting before they were placed on the recipient list. An important element was a psychological consultation with both the potential recipient and the person providing care. No patient was disqualified due to their disability. There was no objection from the patient in any case. Before undergoing transplantation, the patients received outpatient care once a month or per the requirements depending on their health situation. Organ allocation for the study group was carried out on the same terms as for people without disabilities. Of the six patients, four were women, and their median age was 43 (range: 28–61) years, as shown in Table 1. Organs from donors with extended criteria were not transplanted. The average age of the liver donors allocated to the studied group was 35 years. Only one patient died in the postoperative period, as shown in Figure 1. Their caregivers were involved in their postoperative care from the beginning. Medical personnel placed particular emphasis on educating them about the administration of immunosuppressive drugs. As shown in Figure 1, the 1-year survival was 100%; however, the 5-year patient survival rate was 75%. The graft survival rates were statistically similar. Long-term survival was not statistically different from that of the general cohort of electively transplanted patients for chronic liver conditions (1-year survival: 95.8%; 5-year survival: 90.1%). The log-rank tests did not suggest significant differences in the graft survival between groups during the 5-year follow-up (*p* = 0.35). No psychotic incidents were observed in the postoperative period. The median length of hospital stay following transplantation was 24 days and did not differ significantly (*p* = 0.56) from the post-transplantation course of patients without disabilities. No lack of adherence was observed during the post-transplant follow-up visits. The immunosuppressive drug levels (tacrolimus) were within the therapeutic range (5–20 ng/dL), at a median of 12 ng/dL. There were no incidents of rejection of the transplanted livers. The people providing care uniformly acknowledged that transplantation made it easier to care for the patients. None of the patients required a second transplant. All the respondents—the legally entitled caregivers—said they would agree to liver transplantation again if necessary.

## 4. Discussion

Legally incapacitated adults with intellectual disabilities constitute a very minor portion of the patients undergoing total liver transplants, with comparable long-term outcomes to those of adults without disabilities. In such cases, IDDs are not considered barriers to listing on the transplant waiting list or the allocation of an organ. Although there is no national policy for organ allocation leading to transplantation in this patient pool, there is a general assumption that patients without the relevant decisional capacity should have equal access to the waiting list and transplants as long as they satisfy the criteria and indications for liver transplantation and have a structured support system with expected long-term viability. In most cases, the structured support system is a family member, usually a mother. Patients with disabilities cannot be denied healthcare services or access to treatment before they have received an individualized assessment of their general condition. The decision that a patient is not eligible for transplantation is always based on medical evidence, such as a progressive lethal condition or deteriorating general medical status. Nevertheless, one study found that some transplant programs in the USA consider genetic diseases and intellectual disability absolute or relative contraindications to transplantation; adult transplant programs are more likely to do so than children’s [6]. An 11-year-old study of adult liver transplant programs in the USA found that half of the programs considered severe cognitive disability an absolute contraindication to listing for transplantation, versus 1.2% that did for mild cognitive disability [2].

In our experience, strong reinforcement of decision-making regarding access to transplantation for patients with impaired decisional competence comes not just from prior cases of their successful long-term survival but also from general favorable feedback from family or caregivers. Additionally, the existing literature on pediatric populations includes observations of caregivers reporting an improved quality of life (QoL) following transplantation, both for the recipients and the caregivers themselves [1]. To the best of our knowledge, this is the first case series of six highly dependent incapacitated adult patients with IDDs who received liver transplants.

One topic of concern was non-adherence. Non-adherence may be associated with other mental and physical health disorders and could impact long-term survival. However, there are insufficient data to support this statement [7]. In the analyzed group, no non-adherence was reported. Caregivers are responsible for the administration of all medications, including immunosuppressive drugs, since the patients are unable to take care of themselves due to their general condition and legal policy. In our cohort, there was at least one adult caregiver per patient who administered immunosuppressive medication daily. The caregiver was educated on how to provide medicines, how to take care of the patient, and how not to overlook any conditions related to the possible side effects of immunosuppressive drugs. We did not observe any complications during routine checkups or outpatient appointments. Our cohort was found to be more cautious about following the doctor’s directions and medicines than the general population of liver transplant recipients without intellectual disabilities. Favorable results can only be achieved through committed caregivers and social support. Therefore, we believe that liver recipients with strong family support systems are good candidates for liver transplantation.

It should be noted that all the recipients had reached maturity despite having non-treatable medical conditions. The mean age of the patients was 26 years at the time at which this study was conducted. Since childhood, all had been fully dependent on others. For many years, a committed caregiver had been able to meet their basic needs while also managing a variety of coexisting comorbidities. Having a long history of care greatly simplifies decision-making and placing patients on the transplant waiting list. With this support, routine immunosuppressive drug administration is no longer an existential issue for these individuals.

The decision for liver transplant lies in the hands of legal, court-approved caregivers who sign a consent form and are informed of the procedure, its risks, and its benefits. We believe that such a person acts in the best interests of the patient and is fully responsible for the actions taken.

In many families, their financial situation often deteriorates owing to the disability of one of their members, which may hinder access to health services, including the purchase of medicines or routine checkups. The issue of the life expectancy of adult recipients with disabilities is also raised, especially after the death of the assigned caregiver. There are concerns regarding family cooperation, regular intake of immunosuppressive drugs, and the associated risk of losing the transplanted organ [8]. In contrast, it may be argued that over many years of caring for a disabled adult, substantial social and family support networks must be established, particularly in cases where the primary condition has been known about since birth, and committed caregivers act as protectors against all odds. The treatment of potential postoperative complications may be a logistical and legal issue. To reduce the risk of perioperative complications, all transplants were electively planned using grafts of a matched size and weight. Donors with marginal grafts were excluded. To decrease the hospital stress related to transplantation, the family was allowed to be present before and immediately after surgery to comfort the patients and respond to patient-specific needs. This also facilitated communication with the patients. We did not encounter any cases of serious neurological or mental distress during the hospital stays.

There are concerns that patients with IDDs might experience worse outcomes following transplantation due to factors such as a higher risk of complications as a result of the complex transplant procedure. However, research has demonstrated that these concerns are unfounded [8,9]. Growing evidence indicates that individuals with intellectual and developmental disabilities (IDDs) can experience substantial benefits from organ transplantation. Their short- and long-term outcomes, including survival rates, graft survival, and overall quality of life, are comparable to those of individuals without disabilities [1,10,11].

Patients with intellectual disabilities who receive a variety of solid organ transplants have survival rates equal to those without [12,13]. However, none of the relevant studies have focused solely on adult liver transplant recipients. Most of the data have been obtained from children [1,11,14].

In our cohort, graft and patient survival were not inferior to those of the transplant patient population with chronic liver disease but no disabilities. One can argue that such good long-term results are related to selection bias. These were young recipients who had a low model for end-stage liver disease (MELD) score, were generally in good condition, received a selected non-marginal graft, and experienced fewer postoperative complications, with favorable outcomes. Due to the satisfying results of liver transplantation in the present study group, we recommend placing patients with disabilities on the transplant list. However, each patient should be qualified for liver transplantation on an individual basis.

QoL is another concern. The question arises of whether it should be measured arises, and, if so, how to measure it. There is no doubt that the QoL of liver transplantation patients is greatly improved by transplantation [15,16]. For patients with developmental disabilities who are highly dependent and legally incapacitated, QoL is subjective rather than an objective dimension and reflects not only the patient’s status but also the subjective and objective assessments of caregivers on behalf of both parties. Both parties benefit in different ways. Nevertheless, it has been suggested that caregivers should prioritize the quality of patients’ health or care over QoL. This study found that the families were satisfied with the transplant outcomes and expressed willingness to repeat the procedure if needed. None of the patients experienced complications with the immunosuppressive regimens or with the routine inpatient and outpatient clinics.

With appropriate medical care and support, patients with IDDs can safely and effectively undergo transplantation and enjoy the same benefits as those without disabilities. In our experience, some of the benefits that people with IDDs who are fully incapacitated would experience after liver transplantation are as follows: (1) Improved QoL: Transplantation can significantly improve the patient’s physical health; they may experience fewer symptoms of their underlying liver diseases and have fewer medical conditions; (2) improved mobility and independence in some cases; and (3) longer lifespans. Patients with disabilities who receive liver transplants have longer lifespans than those who do not. However, it is important to note that transplantation is not a cure for disability, and it does not eliminate all the challenges faced by patients and caregivers.

Four key transplant outcome measures have emerged in the literature as relevant clinical concerns regarding medical treatment and transplantation for this group of patients [3]. The first is medication adherence, which does not seem to be a major issue provided that there is a committed guardian or caregiver. The next two are patient and graft survival. According to our findings, highly dependent incapacitated patients with IDDs did not have lower liver or patient survival outcomes than other transplant recipients. The last measure is QoL, which does not seem to be worse for the recipients and greatly improves for the caregivers and close family, although further evidence is required to substantiate our findings.

## 5. Conclusions

Highly dependent and legally incapacitated adults with IDDs should not be excluded from the potential pool of liver organ recipients. Their lack of decisional capacity does not affect transplant outcome measures. These patients can safely and effectively undergo liver transplantation. Strong family and social services improve their long-term survival, which is comparable to that of other patients undergoing liver transplantation. Physiological disorders and psychiatric comorbidities in legally incapacitated patients should not prevent them from receiving the benefits of this life-saving and life-enhancing surgery. The decision to perform liver transplantation should be made on a case-by-case basis. Attention should be focused on the caregivers’ wishes and emotions, as they make the final decisions and provide legal consent.

## Figures and Tables

**Figure 1 jcm-13-05702-f001:**
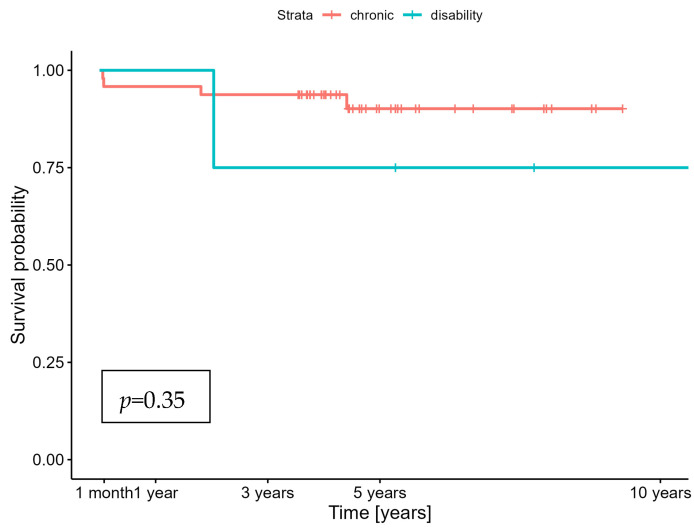
Long-term post-transplant survival among the groups of patients with disabilities and chronic liver failure (CLF) and those without disabilities and chronic liver disease.

**Table 1 jcm-13-05702-t001:** Basic demographic and clinical characteristic of the two groups.

	Patients with Disabilities	Patients with CLF	*p*-Value
Age (years, median)	43	34	0.167
Body mass index (median)	22.57	24	0.399
Total ischaemia time (mean)	478	481	0.670
MELD (median)	28.5	15	0.554
Peak alanine aminotransferase	1213	617	0.069
Peak gammaglutylotransferase	687	687	0.288
Peak alkaline phosphatase	898	1429	0.382

CLF, chronic liver failure; MELD, model for end-stage liver disease.

## Data Availability

The original contributions presented in the study are included in the article, further inquiries can be directed to the corresponding author.

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
