# Peer review of "The Outcomes of Liver Transplantation in Highly Dependent Incapacitated Patients with Intellectual and Developmental Disabilities"

_jcm, 2024, doi:10.3390/jcm13195702_

Round 1

Reviewer 1 Report

Comments and Suggestions for Authors

Thank you for the opportunity to review this interesting paper on an important topic.  This case series is quite small, but as the authors note there is little existing literature and this could be a useful addition. 

General

·      Please consider use of patient centered language rather than impairment centered language e.g. “patient with intellectual disability and incapacity” rather than “incapacitated and disabled patient”

·      There is a considerable number of grammatical errors, typos, and extraneous periods throughout the document

·      I think the discussion section could be shortened considerably.

Abstract

·      Line 27 there appears to be a typo in the 3 year survival statistic.

·      Line 30-31, I agree with this statement, but I don’t understand how it is supported by the results of this study.  The investigators did not report QoL outcomes.

Introduction

·      Line 36-38 please provide a supporting citation

·      Line 40 The Secunda study referenced was a survey of transplant centers performed over 10 years ago.  I think it would be more accurately described as a survey rather than “multicenter  studies.”

·      Line 55-56 While the number of studies and reports are limited for adults with intellectual disabilities who receive transplants, there are pediatric studies.  Consider PMID 33794218, 26655935

·      Line 60—I don’t know what “outline the evidence” means.  I think the authors mean report the outcomes of 6 liver transplants performed in highly dependent, incapacitated adult patients…

Materials and Methods

·      Line 64-66  This paragraph should come after the second paragraph.  Please provide a citation for the definition of incapacity used.  Each country has different processes and statuetes.  Line 66-69—is altered consciousness part of the description of 6 recipients?  It should be with the remainder of the description.  Also altered consciousness is a different description than incapacitated, dependent, or disabled—patients can be all of those things without altered consciousness.

·      Where were these transplants performed?  At Warsaw Medical University?  In one country?

·      Line 75—how could informed consent be obtained from all subjects involved in the study? 

Results

·      The results section needs to be expanded—the authors make several references to results in the discussion that are not included in the results section.

·      Line 93-94—If a patient died postoperatively, how is the 1 year survival 100%?

Discussion

·      Line 130-132—I don’t think the referenced citation supports the claims about improved QoL.

·      Line 132-133—Who are the respondents?  If it is part of this study it needs to be reported in the results section

·      Line 136—Who was concerned about nonadherence?  Why?  Intellectual disability is not a mental health disorder.

·       Line 148-149—I would not that not all in transplant agree with this assertion.  See for example PMID 31647757

·      Line 189-190—I disagree that patients with ID who receive transplants have survival rates equal to those without.  They have survival rates that are similar.

·      Line 212-222—This paragraph should be moved to the results section if Qol, mobility, and independence were assessed or removed entirely.

Comments on the Quality of English Language

Numerous grammatical errors and extraneous periods and commas

Reviewer 2 Report

Comments and Suggestions for Authors

your article highlights a specific problem. However, transplanting underage children is no different than transplanting patients with intellectual and developmental disabilities. The key is to have a good network around the patient. What is sometimes an issue, is that the caregivers are the elderly parent(s) of the patient. When they are passing away, no one replaces them.

But indeed having intellectual or developmental disabilities doesn't preclude transplantation. Everybody has the right to live a good life.

some remarks:

p1 263/5 y survival: 9,7%?

p3 95: 75 % 5 y survival rate: explanation

p5 195: low meld: this is not the case in your population.Explain

p3 table 1: high meld scores of pts with disabilities: explanation?

you should perhaps stress more the importance of the aftercare in your discussion

Round 2

Reviewer 1 Report

Comments and Suggestions for Authors

Thank you for the opportunity to again review this paper.  This revision is a significant improvement. 

Results: Line 96-97. This study reported the outcomes of liver transplant recipients-- it does not allow for direct comment on who is on transplant waiting lists.  It is more precise to note that highly dependent and incapacitated transplant patients form an extreme minority of liver transplant recipients.  This is important, because this study and others of transplant recipients cannot describe those with IDDS and organ failure never referred for transplant, those referred and declined, or those referred and accepted but remain on the wait list or die on the wait list.

Line 99-100-- why was psychological consultation important?  Why is a description of psychological consultation and its importance part of the results section rather than the methods or discussion?

Line 119-122-- was the data on caregiver perception of care after transplant and preference about retransplant accessible in the retrospective review of the medical record?  If not, please describe how these perspectives were obtained.

Discussion 

line-139-140. I disagree with the assertion that "the decision that a patient is not eligible for transplantation is always based on medical evidence."  I think there is extensive published data and cases within the public domain where transplant centers did not reach this standard.  If the authors are referring to the process at their center this should be clarified.  If the authors are instead asserting that "the decision that a patient is not eligible for transplantation SHOULD always based on medical evidence" this should be clarified and further supported with explanation or references.

Line 171-- the mean age of 26 years at time study was conducted appears to contradict the median age of recipients in table 1 (46 years).  Please clarify

Line 203-- the three cited studies did not report quality of life.

Line 222-223: I don't understand what the sentence beginning with "nevertheless" means.  Please also add a citation for who has made the "suggestion"

Minor: Reference 1 and 12 are the same study

Comments on the Quality of English Language

English has been improved

Author Response

Abstract

  • Line 27 there appears to be a typo in the 3 year survival statistic- I made the correction;
  • Line 30-31, I agree with this statement, but I don’t understand how it is supported by the results of this study.  The investigators did not report QoL outcomes- we clarified this statement

Introduction

  • Line 36-38 please provide a supporting citation- I provided a supporting citation (please see ref [1])
  • Line 40 The Secunda study referenced was a survey of transplant centers performed over 10 years ago.  I think it would be more accurately described as a survey rather than “multicenter  studies.”- I agree; we changed it to survey
  • Line 55-56 While the number of studies and reports are limited for adults with intellectual disabilities who receive transplants, there are pediatric studies.  Consider PMID 33794218, 26655935- considered these two studies in pediatric group
  • Line 60—I don’t know what “outline the evidence” means.  I think the authors mean report the outcomes of 6 liver transplants performed in highly dependent, incapacitated adult patients…- we changed to ‘report’

Materials and Methods

  • Line 64-66  This paragraph should come after the second paragraph.  Please provide a citation for the definition of incapacity used.  Each country has different processes and statuetes.  Line 66-69—is altered consciousness part of the description of 6 recipients?  It should be with the remainder of the description.  Also altered consciousness is a different description than incapacitated, dependent, or disabled—patients can be all of those things without altered consciousness- we changed as suggested, please see in 64-69 lines
  • Where were these transplants performed?  At Warsaw Medical University?  In one country- yes the Department Of General, Transplant and Liver Surgery of Public Central Teaching Hospital, Medical University of Warsaw
  • Line 75—how could informed consent be obtained from all subjects involved in the study- good suggestion

Results

  • The results section needs to be expanded—the authors make several references to results in the discussion that are not included in the results section. – We expanded the results section, for more details please see lines 96-122
  • Line 93-94—If a patient died postoperatively, how is the 1 year survival 100% It was mistakingly assumed that this patient died postoperatively, in fact this patient died after 1 year post liver transplantation

Discussion

  • Line 130-132—I don’t think the referenced citation supports the claims about improved QoL- that’s true, we modifies this, please see lines 130-132
  • Line 132-133—Who are the respondents?  If it is part of this study it needs to be reported in the results section; we reported as suggested
  • Line 136—Who was concerned about nonadherence?  Why?  Intellectual disability is not a mental health disorder- we modifies also this line
  • Line 148-149—I would not that not all in transplant agree with this assertion.  See for example PMID 31647757; we agree please see in the text
  • Line 189-190—I disagree that patients with ID who receive transplants have survival rates equalto those without.  They have survival rates that are similar- that’s right, we modified this as suggested
  • Line 212-222—This paragraph should be moved to the results section if Qol, mobility, and independence were assessed or removed entirely- we incorporated this to results
